# Pancreatic Neuroendocrine Tumors: What Is the Best Surgical Option?

**DOI:** 10.3390/jcm13103015

**Published:** 2024-05-20

**Authors:** Renato Patrone, Federico Maria Mongardini, Alessandra Conzo, Chiara Cacciatore, Giovanni Cozzolino, Antonio Catauro, Eduardo Lanza, Francesco Izzo, Andrea Belli, Raffaele Palaia, Luigi Flagiello, Ferdinando De Vita, Ludovico Docimo, Giovanni Conzo

**Affiliations:** 1Hepatobiliary Surgical Oncology Unit, Istituto Nazionale Tumori IRCCS Fondazione Pascale-IRCCS di Napoli, 80131 Naples, Italy; f.izzo@istitutotumori.na.it (F.I.); a.belli@sititutotumori.na.it (A.B.); r.palaia@istitutotumori.na.it (R.P.); ludovico.docimo@unicampania.it (L.D.); giovanni91cozzolino@libero.it (G.C.); 2Division of General, Oncological, Mini-Invasive and Obesity Surgery, University of Study of Campania “Luigi Vanvitelli”, 80138 Naples, Italy; federicomaria.mongardini@unicampania.it (F.M.M.); aleconzo@hotmail.it (A.C.); cacciatore.chiara@virgilio.it (C.C.); giovanni.conzo@unicampania.it (G.C.); antonio.catauro@unicampania.it (A.C.); eduardo.lanza@libero.it (E.L.); luigi.flagiello@studenti.unicampania.it (L.F.); 3Division of Medical Oncology, Department of Internal and Experimental Medicine ‘F. Magrassi’, Università della Campania ‘Luigi Vanvitelli’, 80138 Naples, Italy; ferdinando.devita@unicampania.it

**Keywords:** pNET, neuroendocrine tumors, pancreas tumor, pancreatic cancer, enucleation, pancreatectomy

## Abstract

**Background**: Pancreatic neuroendocrine tumors (pNETs) represent a rare subset of pancreatic cancer. Functional tumors cause hormonal changes and clinical syndromes, while non-functional ones are often diagnosed late. Surgical management needs multidisciplinary planning, involving enucleation, distal pancreatectomy with or without spleen preservation, central pancreatectomy, pancreaticoduodenectomy or total pancreatectomy. Minimally invasive approaches have increased in the last decade compared to the open technique. The aim of this study was to analyze the current diagnostic and surgical trends for pNETs, to identify better interventions and their outcomes. **Methods**: The study adhered to the PRISMA guidelines, conducting a systematic review of the literature from May 2008 to March 2022 across multiple databases. Several combinations of keywords were used (“NET”, “pancreatic”, “surgery”, “laparoscopic”, “minimally invasive”, “robotic”, “enucleation”, “parenchyma sparing”) and relevant article references were manually checked. The manuscript quality was evaluated. **Results**: The study screened 3867 manuscripts and twelve studies were selected, primarily from Italy, the United States, and China. A total of 7767 surgically treated patients were collected from 160 included centers. The mean age was 56.3 y.o. Enucleation (EN) and distal pancreatectomy (DP) were the most commonly performed surgeries and represented 43.4% and 38.6% of the total interventions, respectively. Pancreatic fistulae, postoperative bleeding, re-operation, and follow-up were recorded and analyzed. **Conclusions**: Enucleation shows better postoperative outcomes and lower mortality rates compared to pancreaticoduodenectomy (PD) or distal pancreatectomy (DP), despite the similar risks of postoperative pancreatic fistulae (POPF). DP is preferred over enucleation for the pancreas body–tail, while laparoscopic enucleation is better for head pNETs.

## 1. Introduction

Pancreatic neuroendocrine tumors (pNETs) were first described in 1869 as a rare subset of neuroendocrine neoplasms (NENs) [1]. They have a higher prevalence in Caucasian patients (84%) and the male population and their incidence increases with age [2]. Tumors can be defined as functional if they overproduce hormones that result in a distinct clinical syndrome; others are non-functional if they show no or minimal secretion of hormones, without resulting in a manifested syndrome, and 90% of them are silent and diagnosed in the late stages [3,4]. Rarely, tumors are included in hereditary syndromes such as multiple endocrine neoplasia type 1 (MEN-1), Von Hipple–Lindau (VHL) disease, and neurofibromatosis type 1 (NF-1), while they generally occur in a sporadic form [5]. The development of diagnostic tools and the whole genome landscape has led to significant growth in the impact of NF-pNETs in the last 20 years [6].

The surgical management of pNETs should be planned in a multidisciplinary staff meeting, including accurate preoperative staging (localization and grading) [7]. Parenchyma-sparing surgery should be performed for asymptomatic NF-pNETs > 2 cm and all functional sporadic pNETs except those with unresectable distant metastases [8]. In contrast, according to the Italian Medical Oncology Association’s guidelines for NENs, a clinical and radiological follow-up is recommended in the case of a non-functioning pNET < 2 cm [9]. Open or minimally invasive surgical treatment involves enucleation, distal pancreatectomy with or without spleen preservation, and central pancreatectomy; pancreaticoduodenectomy (PD) or total pancreatectomy may be indicated in specific cases [10]. In the last two decades, the surprising diffusion of minimally invasive surgery has introduced a new scenario modifying the surgical approach, reducing the number of enucleation procedures in favor of distal pancreatectomy, which is considered safer and easier from a technical point of view.

Due to this wide variability in surgical options, the authors analyzed the current trends in terms of diagnosis and surgical approaches in the treatment of pNETs, with the aim to identify the preferred intervention and medium–long-term results.

## 2. Materials and Methods

The present study was accomplished in accordance with the Preferred Reporting Items for Systematic Reviews and Meta-Analyses (PRISMA) guidelines [11]. A systematic review of the studies published in the literature from May 2008 to March 2022 was independently performed by two authors (F.M.M., G.COZ.) in 4 databases (PubMed, Scopus, Google Scholar, and Medline), based on criteria predetermined by the investigators. The obtained results were discussed with the senior authors (R.P. and G.C.O.N). The research included not only original articles, meta-analyses, and reviews, but also the Cochrane database and textbooks, whose citations were further cross-checked. Discrepancies in data collection, classification, and analysis were resolved by the consensus of all authors. Several combinations of the keywords and Medical Subject Heading (MeSH) search terms were used, including “NET”, “pancreatic”, “surgery”, “laparoscopic”, “minimally invasive”, “robotic”, “enucleation”, and “parenchyma-sparing”. The various terms were included consecutively during the search. The references of the more relevant articles were manually searched.

A pNET retrospective review analyzing epidemiology data, diagnosis, surgical approaches, and the medium–long-term results of surgical treatment was performed. The authors, considering the main literature collections of pNET patients, evaluated the current clinical trends in light of modern knowledge and of the diffusion of minimally invasive surgery (MIS).

The following inclusion and exclusion criteria were applied.

Inclusion criteria:(1)English language studies including patients with a clinical diagnosis of a pNET;(2)Open surgery vs. minimally invasive surgery (laparoscopic or robotic) for the surgical treatment of pNETs;(3)Surgical treatment related to pNET localization reporting at least one intraoperative, perioperative, or postoperative outcome.

Exclusion criteria:(1)Non-English studies;(2)Animal studies;(3)Non-comparative studies;(4)Abstracts, expert opinions, editorials, and letters to the editor;(5)Studies reporting inadequate clinical data;(6)The treatment of other pancreatic tumors.

All studies that failed to fulfil the established inclusion criteria were automatically rejected.

The evaluation of the manuscript quality was conducted using the Methodological Index for Non-Randomized Studies (MINORS) criteria [12] and the Newcastle–Ottawa Scale (NOS) [13] to assess the quality of the non-randomized studies in meta-analyses because of the non-randomized nature of the selected papers. 

## 3. Results

### 3.1. Evaluation and Inclusion

We found a total of 3867 manuscripts for the initial screening, using systematic research. At the first step, we excluded 1513 papers because they appeared as duplicates. Analyzing the titles, abstracts, and keywords, the authors selected the full-text versions of 43 papers. The main reasons for exclusion were the absence of patients treated both with MIS and open approaches (n = 478), no surgical patients (n = 216), and the inclusion of other types of pancreatic cancer (n = 401). Further causes of exclusion were populations treated with palliative intent or case series or the absence of specific data on the postoperative outcomes. Another 19 studies were excluded during the full-text examinations due to the inclusion of other benign pancreatic lesions and a lack of systematic postoperative data. Moreover, nine articles were excluded during data extraction because of inadequate clinical data. This led to the final selection of 12 studies that fulfilled the inclusion criteria [10,14,15,16,17,18,19,20,21,22,23,24].

The search strategy flow diagram is shown in Figure 1.

No randomized clinical studies were found. All selected papers were retrospective studies and five of them were multicenter studies [10,15,16,17,20]. The geographical distribution of the selected papers was as follows: two from Italy, the United States, and China; one from Germany (1), Spain, Romania, the Netherlands, Korea, and Norway.

The characteristics of the included manuscripts are summarized in Table 1.

The process of the quality evaluation, following the MINORS and NOS criteria, is reported in Table 2 [12,13].

### 3.2. Baseline Characteristics

A total of 7767 surgically treated patients were collected from 160 included centers. The mean age was 56.3 y.o. (± 2.65 SD), with a prevalence of the female sex (57.4%) (Table 1).

All patients had a histopathological diagnosis of pNET on surgical specimens.

### 3.3. Surgical and Postoperative Results

Enucleation (EN) and distal pancreatectomy (DP) were the most commonly performed surgeries and represented 43.4% and 38.6% of the total interventions, respectively. Other reported surgeries were pancreaticoduodenectomy (PD) at 8%, central pancreatectomy (CP) at 1.1%, total pancreatectomy (TP) at 0.8%, and various other surgical strategies, e.g., multivisceral resection, at 7.9% (Table 3).

A pancreatic fistula (PF) was described in 10 of the 12 selected papers and was reported in 15.8% of patients (regardless of the type of surgical intervention) (Table 4).

Postoperative bleeding (POB) occurred in 160 patients and the mean length of hospital stay (LHS) was 6.7 days (Table 5).

Re-operation was needed in 156 patients (1.77%) with a 3% rate of 30-day postoperative mortality.

The mean follow-up duration was 41.5 months.

## 4. Discussion

NETs represent a cluster of complex oncological malignancies originating from neuroendocrine cells, with a large range of clinical presentations and anatomical localizations, especially in the digestive tract, the lungs, and the pancreas [1]. Although pNETs represent 1–2% of all pancreatic neoplasms, their incidence is increasing [25]. Their diagnosis is usually complex and requires blood sampling for chromogranin A and synaptophysin [26,27] and a specific search for hormones such as serotonin, gastrin, insulin, and glucagon, to confirm the diagnosis in patients with clinical symptoms. These remain the main diagnostic tools useful to obtain accurate localization, grading, and staging in pNET management. In the last few years, gallium scintigraphy and positron emission tomography (PET) scans have been introduced but are sometimes overutilized for functional pre- and postoperative studies. Ga 68 DODATOC PET/TC has better results regarding sensitivity than an octreoscan (90–100% vs. 50–80%), especially for the identification of liver micrometastases and locoregional lymph nodes [27].

As regards intraoperative localization, ecolaparoscopy is recommended in every case, with the aim of achieving better tumor identification, a better evaluation of the healthy pancreatic parenchyma, and a better lymph node status. This diagnostic tool helps the surgeon to make the correct decision regarding surgical intervention: the precise identification of the relationship between the tumor and the pancreatic duct is crucial in choosing the surgical procedure (enucleation vs. distal pancreatectomy) according to the European Neuroendocrine Tumor Society (ENETS) and the AIOM guidelines on PS [8,9,28].

Table 1 presents a detailed overview of the surgical procedures and patient demographics across various international centers regarding the treatment of pNETs. A total of 8488 patients were included in the analysis, with a predominance of females (53.4%) and an average age of 58.3 years. This demographic profile aligns with the literature characteristics of pNET patients, who are often diagnosed in their fifth to sixth decades of life [29].

Table 2 offers a comprehensive analysis of the patient demographics, surgical approaches, and postoperative outcomes in the treatment of pancreatic neuroendocrine tumors (pNETs) across multiple studies and centers worldwide. The demographic data reveal interesting trends in the distribution of patients. While the female-to-male ratio varies slightly across studies, with some showing a higher proportion of female patients, the mean age of patients is generally clustered around 58.3 years, indicating that pNETs commonly affect individuals in their late fifties.

In most cases, following a multidisciplinary evaluation, a surgical approach was considered. The surgical strategy was related to the volume, the number, and the localization of the lesions and their distance to the pancreatic duct. All patients underwent surgical laparotomy or laparoscopy in order to remove the pNETs and to analyze them histologically. Open, laparoscopic, or robotic surgical choices involve enucleation, distal pancreatectomy (DP) with or without splenectomy, central pancreatectomy, pancreaticoduodenal resection (Whipple’s operation) with or without pylorus preservation, and total pancreatectomy according to the primary tumor location [30,31,32].

The patient characteristics, tumor histomorphology, size and location, surgeon preferences, and local availability of resources are the preferred decision-making factors.

The main focus of surveys on I-GO-MIPS (a minimally invasive pancreatic surgery), according to Zerbi et al., is to analyze the volume, type, and surgical indications for minimally invasive pancreatic surgery to assess the postoperative outcomes and increase the use of MIPS in Italian centers and to contribute to the European registry [33,34]. In the last two decades, the rate of pNET surgery has been improved, especially regarding minimally invasive distal pancreatic surgery on the pancreatic body and tail [35,36]. The current preference of parenchyma-sparing techniques such as enucleation and middle pancreatectomy (for lesions of the central portion of the gland) arises from the high risk of pancreatic endocrine/exocrine insufficiency following PD and DP [37].

Surgical interventions exhibited notable disparities in the frequency and type of procedure performed across different centers. Enucleation (EN), a procedure aimed at preserving the pancreatic tissue, emerged as a prominent choice in several studies, particularly in centers located in Italy and China. This trend may reflect a growing emphasis on preserving pancreatic function and minimizing surgical invasiveness. Conversely, more extensive procedures such as pancreaticoduodenectomy (PD) and total pancreatectomy (TP) were performed less frequently but were still significant in managing advanced or complex cases.

Enucleation (EN) represents the most common surgical treatment for pNETs (43.4%) and was reported, as a surgical treatment, by nine authors [10,14,15,16,18,19,20,21], but Zerbi et al. did not divided their cases into minimally invasive and open approaches [16]. The surgical interventions varied across centers, with enucleation (EN) being the most common procedure, accounting for 43.4% of cases (Table 1). Enucleation is favored for small, well-defined tumors, as it allows for tumor removal while preserving the pancreatic parenchyma. Despite the large diffusion of laparoscopy and, in the few last years, of robotic surgery, EN was performed in open surgery in 89.5% and was reported at 7.3% of conversion in MIS approaches.

In the series by Haugvik et al., the laparoscopic enucleation of pNETs of the pancreatic head was preferred over major surgery (Whipple’s procedure); on the other hand, the laparoscopic resection of pNETs in the pancreatic body and/or tail could be favored over enucleation [19]. This preference could be justified by the lower invasiveness and potential benefits of laparoscopy, such as reduced blood loss, shorter hospital stays, and quicker postoperative recovery, compared to the Whipple procedure, involving more extensive pancreatic and intestinal resection. On the other hand, for pNETs located in the body and/or tail of the pancreas, laparoscopy might be preferred over enucleation. The concept of the safety of parenchyma sparing has been highlighted in relation to the number of postoperative deaths after PD [38]. In summary, the choice between enucleation and laparoscopic resection depends on the tumor’s location and specific characteristics, as well as the consideration of the long-term complications and benefits for the patient.

According to Beane et al., the postoperative outcome comparison between enucleation and resection showed a higher PD or DP mortality rate than enucleation (*p* = 0.016). Meanwhile, the rates of serious morbidities, lengths of stay, and surgical site infections were lower for both DP and enucleation than PD [39]. According to Jilesen et al., postoperative morbidity after the enucleation of pNETs was comparable to that of PD or DP and therefore it must be considered a high-risk operation. In every case, pNETs need a preoperative evaluation to determine the appropriate size, location, and functional status [20].

The long-term oncological outcomes are similar between LDP for pNETs and ODP, with decreasing total morbidity in terms of postoperative complications without compromising survival [40]

Distal pancreatectomy (DP) was the second most frequently performed procedure, representing 38.6% of cases (Table 1). DP is typically indicated for tumors located in the body or tail of the pancreas and offers a curative option for localized disease. Moreover, in this case, the laparoscopic approach was reserved only for a small segment of the population, with 21.2%.

Regional variations in surgical practices were evident, with centers in Europe and Asia demonstrating distinct preferences. For instance, Italian centers favored EN, while Norwegian and Dutch centers adopted a more balanced approach between EN and distal pancreatectomy (DP). In contrast, centers in the United States showed the greater utilization of DP, possibly influenced by local surgical expertise and institutional protocols. Temporal trends revealed a gradual shift towards minimally invasive techniques, with laparoscopic and robotic-assisted surgeries gaining traction in recent years. While these approaches offer advantages such as reduced postoperative pain and shorter hospital stays, their adoption varied across regions, reflecting differences in technological infrastructure and surgeon training.

Nevertheless, the MIS approach represented a more frequent surgical option in SDP than EN. These unexpected results, in our opinion, could be due to the better standardization of SDP than EN, which represents a surgically tailored intervention based on the specific needs of the patient and often requires ultrasonography for the correct identification of the lesions and for a better evaluation of its distance from the main duct. Another possible explanation for this result could be that, if the surgeon is unsure about performing EN, they might prefer the open approach, because it would be simpler to modify the surgical strategy in order to perform a more complex surgical intervention (TP, CP, or DP). In fact, pancreaticoduodenectomy (PD), which accounts for 8% of all surgical procedures, as well as total pancreatectomy (TP) at 1.1% and central pancreatectomy (CP) at 0.8%, exhibited notably low rates regarding minimally invasive approaches, with only 2.7%, 1.1%, and 1%, respectively. This indicates that these complex procedures, which involve the extensive resection of the pancreatic tissue and intricate anatomical reconstruction, are less frequently performed using minimally invasive techniques such as laparoscopy or robotics. This is obviously related to the high surgical complexity, but there were also specific study limitations: our selected study collected data from papers published since 2008 [21]; only six papers were published after 2015 [14,15,17,20,22,23] (Table 2).

Five authors reported various other surgical interventions (e.g., multivisceral resections, atypical pancreatic resection) [10,16,18,19,21] (Table 2).

Data from the literature clearly explain that MIS for pancreatic cancer has seen rapid development in the past decade, and, since 2017, promising outcomes have been reported by early adopters from high-volume centers [15]. Subsequently, multicenter series as well as randomized controlled trials have been reported, with increasing scientific publications in this field [41]. This paper considered the MIS approach to all pancreatic cancers, but our focus was on pNETs, a specific and rare disease; for this reason, it was not possible to select the most up-to-date papers, with the exception of Sutton et al. [15]. The safety and efficacy of laparoscopic and open surgical approaches is compared in a number of studies, and improved results were reported following laparoscopic procedures in terms of cosmetic scars, reduced postoperative pain, and hospitalization [42]. The laparoscopic technique speeds up the oral intake after surgery and the recovery of gastrointestinal function [43]; its use is greatly increased in several pancreatic diseases, despite the complexity due to the long operating time and the long learning curve. In the series by Casadei et al., 14.6% of patients needed distal pancreatectomy, particularly those with non-malignant pancreatic tumors (90.9%), who underwent laparoscopic distal pancreatectomy associated with splenectomy and rarely with spleen preservation [24].

Particular attention should be reserved for robotic pancreatic resection or enucleation, and several advantages are connected to specific functions. Clinical series are lacking and they are derived from referral centers, where the surgical pancreatic experience is associated with advanced laparoscopic skills. More data are necessary to confirm that a robotic surgical approach is a safe and viable procedure [44]. Nonetheless, this surgical treatment offers considerable advantages, such as a lower conversion rate, parenchyma sparing, and a reduction in hospitalization, despite the high costs and longer operative times. The literature shows a significant lack of data on robotic pNET surgery, so a comparison of other surgical techniques and robotic surgery is not feasible. It is possible that the oncological and perioperative scores between the laparoscopic and robotic approaches are comparable, as evidence-based medicine (EBM) describes [45].

Regarding the postoperative outcomes, the pancreatic fistula rate, length of hospital stay (LHS), and blood loss were evaluated. Regarding the post-surgical results, postoperative bleeding (POB) with a Clavien–Dindo classification score ≥ 2 was reported in 2% of all patients, with a median overall length hospital stay (LHS) of 6.75 days [46]. Moreover, 156 patients needed readmission to the surgical room: 45 for pancreatic fistulae, 87 for POB, and 24 for other reasons (perforations, anastomotic leak).

The incidence of pancreatic fistulae varied across studies, with some demonstrating higher rates compared to others. The classification of pancreatic fistulae into Grades A, B, and C (following the International Study Group on Pancreatic Fistulae—ISGPF—criteria) offers crucial insights into the severity and management of postoperative pancreatic complications [47]. According to our analysis, a higher rate of POPF is most often related to the laparoscopic enucleation of pNET than laparoscopic resection. Fernandez-Cruz et al. reported a pancreatic fistula rate of 8%, while Haugvik et al. observed a lower rate of 7% [19,21]. These differences may be attributed to variations in the patient selection criteria, surgical techniques, and postoperative management protocols. Notably, Zhao et al. and Zhang et al. reported relatively higher rates of pancreatic fistulae, suggesting potential challenges or complexities associated with performing pancreatic surgery in these centers [17,18]. Possible explanations for these challenges could include variations in the surgical techniques, the expertise levels among surgical teams, patient populations with more advanced disease or comorbidities, differences in postoperative management protocols, or institutional factors affecting the quality of care. This result underscores the importance of ongoing quality improvement efforts, the standardization of surgical protocols, and continued education and training for surgical teams to optimize the outcomes and minimize complications in pancreatic surgery.

Studies employing enucleation (EN) as the primary surgical approach reported higher rates of PF compared to more extensive procedures like pancreaticoduodenectomy (PD) and total pancreatectomy (TP). This underscores the inherent challenges associated with preserving the pancreatic tissue’s integrity during surgery. While EN offers advantages in terms of preserving pancreatic function and reducing postoperative morbidity, it also carries a higher risk of PF formation due to the delicate nature of tissue dissection and manipulation. Conversely, PD and TP, although associated with lower PF rates, entail greater pancreatic resection and may result in exocrine and endocrine insufficiency. The variation in the PF rates underscores the importance of tailored surgical decision-making based on the tumor characteristics, patient comorbidities, and surgical expertise.

The LHS is a critical parameter in assessing the efficiency of surgical management and postoperative recovery. There is considerable variability in the LHS across different centers, reflecting differences in patient populations, surgical techniques, and institutional practices. Notably, centers with a high minimally invasive approach rate, such as laparoscopic and robotic-assisted surgeries, tended to exhibit a shorter LHS compared to those using traditional open procedures. This result highlights the potential benefits of minimally invasive techniques in facilitating faster postoperative recovery and reducing healthcare resource utilization. Casadei et al. and Mehrabi et al. reported a shorter median LHS, indicating efficient postoperative care and patient recovery [10,24]. Conversely, Zhang et al. and Xourafas et al. showed a longer LHS, possibly influenced by factors such as postoperative complications and institutional protocols [17,22]. The range of LHS values across studies underscores the importance of identifying the factors contributing to prolonged hospital stays and of implementing strategies to optimize perioperative care and discharge planning. However, it is essential to interpret LHS data in conjunction with other clinical factors, such as complication rates and patient outcomes, to assess the overall quality and safety of surgical care.

Blood loss during pancreatic surgery can impact postoperative outcomes and necessitate transfusions, thereby increasing the risk of complications. The analysis revealed variations in intraoperative blood loss across different procedures and centers. Zhao et al. and Zhang et al. reported varying degrees of blood loss, with median values ranging from 149 mL to 716.7 mL [17,18]. These differences may reflect variations in surgical complexity, tumor characteristics, and intraoperative management. While studies implementing minimally invasive techniques reported lower intraoperative blood loss compared to traditional open procedures, the magnitude of the reduction varied depending on the complexity of the surgery and the surgeon’s proficiency in minimally invasive approaches. Effective intraoperative hemostasis techniques, meticulous surgical planning, and advanced surgical technologies such as intraoperative blood salvage systems contribute to minimizing blood loss and optimizing patient outcomes.

The postoperative outcomes provide valuable insights into the safety and efficacy of surgical interventions. While EN was associated with shorter hospital stays compared to more extensive procedures like PD and TP, it also exhibited higher rates of pancreatic fistula formation, highlighting the importance of careful patient and surgical technique selection. Additionally, the differences in the postoperative bleeding rates underscore the need for standardized perioperative management protocols to mitigate complications and optimize patient outcomes.

In summary, the analysis of the pancreatic fistula rate, length of hospital stay, and blood loss across different studies provides valuable insights into the variability and determinants of postoperative outcomes in pancreatic surgery. Future research endeavors should focus on elucidating the factors contributing to these outcomes and developing standardized protocols to optimize surgical care and enhance patient recovery.

Despite the comprehensive nature of the dataset, several limitations should be acknowledged. The retrospective design of the studies may introduce selection bias, and the heterogeneity in reporting outcomes across centers hinders direct comparisons. The authors used data from published studies but could not ensure their consistency and reliability. Additionally, the absence of randomized controlled trials limited the ability to draw definitive conclusions regarding the optimal surgical approach for pNETs.

## 5. Conclusions

In our analysis, enucleation is associated with better rates in terms of postoperative outcomes and mortality, despite having the same risk of POPF as PD or DP. DP should be favored among the minimally invasive procedures for surgery on the pancreatic body–tail as compared to enucleation; on the other hand, laparoscopic enucleation is performed in head pNETs. In conclusion, the comprehensive analysis of the demographic characteristics, surgical practices, and postoperative outcomes elucidates the multifaceted nature of pNET management. Moving forward, efforts to standardize surgical protocols, enhance training in minimally invasive techniques, and promote multidisciplinary collaboration are crucial in improving the quality of care and optimizing the outcomes for patients with pNETs. Furthermore, our review shows that preoperative multidisciplinary management is necessary to achieve better pancreatic treatment, and the patient’s characteristics, tumor localization, functional state, and tumor size are important decision-making factors regarding surgery. In this direction, more comprehensive international guidelines are needed. In our opinion, the diffusion of the robotic approach, with its well-known and undisputed advantages, will allow minimally invasive surgery in more centers. In the near future, all pNETs of the tail or body will be managed with minimally invasive and anatomical surgery, abandoning enucleation.

## Figures and Tables

**Figure 1 jcm-13-03015-f001:**
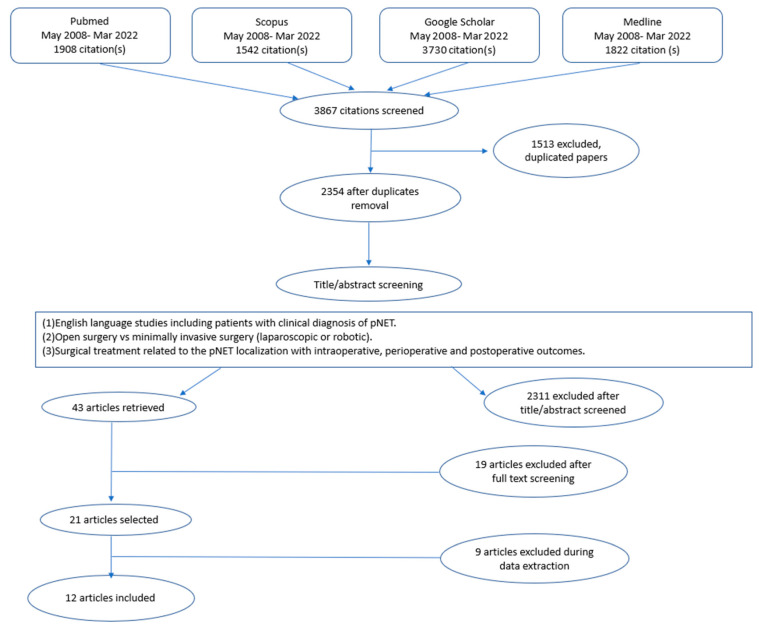
PRISMA flowchart. Diagram of the systematic review of the literature, performed in 4 databases from May 2007 to March 2022. Search terms included “NET”, “pancreatic”, “surgery”, “laparoscopic”, “minimally invasive”, “robotic”, “enucleation” “parenchyma-sparing”. Inclusion criteria are shown in the central box. Major reasons for exclusion were the presence of a clinical diagnosis, localization, and grading of pNET (n= 475) and the intraoperative, perioperative, and postoperative outcomes of pNET surgery (open or minimally invasive) (n = 1836). Further reasons for exclusion were non-comparative studies and those reporting inadequate or partial clinical data. This led to the final selection of 13 studies that fulfilled the inclusion criteria.

**Table 1 jcm-13-03015-t001:** A population characteristic summary of the selected studies [10,14,15,16,17,18,19,20,21,22,23,24].

		Sex		Age (Years)			N. Center
Author	N. Pts	M	F	Mean Age	Max	Min	
Fernandez-Cruz	49	6	43	58	83	22	1
Casadei	44	8	36	60.5	77	42	1
Zerbi	310	166	144	57.6			24
Zhao	328	139	189	42.3	57	27	1
Haugvik	65	27	38	57	87	21	1
Mehrabi	6222	2551	3671	53	72	27	114
Xourafas	171	89	82	61.5	95	20	1
Jilesen	205	93	112	52.6	68	34	2
Han	94	55	39	53.5	75	30	1
Mihalache	18	6	12	53	69	28	1
Zhang	1020	464	556	58	66	48	9
Sutton	282	142	140	59	77	42	4

**Table 2 jcm-13-03015-t002:** Quality evaluation following Methodological Index for Non-Randomized Studies (MINORS) and Newcastle–Ottawa Scale (NOS) criteria—ProS: prospective study; RetS: retrospective study, SC: single center, MC: multi-center, DB: database, *: low quality, **: medium quality, ***: high quality [10,14,15,16,17,18,19,20,21,22,23,24].

Year	Author	Country	Type of Study	N. Center	NOS			MINORS
					Selection	Comparability	Outcome/Exposure	
2008	Fernandez-Cruz	Spain	ProS-SC	1	**	*	**	15
2010	Casadei	Italy	ProS-SC	1	***	**	***	22
2011	Zerbi	Italy	ProS-MC	24	***	*	**	15
2011	Zhao	China	RetS-SC	1	***	**	**	18
2013	Haugvik	Norway	RetS-SC	1	**	**	**	16
2014	Mehrabi	Germany	RetS-DB	114	***	**	***	20
2015	Xourafas	USA	RetS-SC	1	**	**	***	18
2015	Jilesen	Netherlands	RetS-MC	2	***	**	***	19
2017	Han	Korea	RetS-SC	1	***	**	***	20
2019	Mihalache	Romania	RetS-SC	1	**	*	**	16
2019	Zhang	China	RetS-MC	9	***	**	***	19
2022	Sutton	USA	RetS-MC	4	***	*	***	18

**Table 3 jcm-13-03015-t003:** All surgical interventions reported, divided by author. Values are reported for the overall rate, open surgery, laparoscopic (LAP) approach, and conversion rate (CR) [10,14,15,16,17,18,19,20,21,22,23,24].

	Enucleation				Splenodistal Pancreatectomy				Pancreaticoduodenectomy				Central Pancreatectomy				Total Pancreatectomies			
Author	Overall	Open	Lap	CR	Overall	Open	Lap	CR	Overall	Open	LAP	CR	Overall	Open	Lap	CR	Overall	Open	Lap	CR
Fernandez-Cruz	22	0	21	1	26	0	23	3	\	\	\	\	\	\	\	\	\	\	\	\
Casadei	\	\	\	\	44	22	22	0	\	\	\	\	\	\	\	\	\	\	\	\
Zerbi	50	\	\	\	114	\	\	\	55	\	\	\	16	\	\	\	12	\	\	\
Zhao	229	199	18	12	53	37	9	7	3	3	0	0	15	15	0	0	\	\	\	\
Haugvik	16	0	14	2	53	0	51	2	\	\	\	\	\	\	\	\	\	\	\	\
Mehrabi	2866	2677	189		1603	1506	97		140	139	1	0	19	19	0	0	29	29	0	0
Xourafas	\	\	\	\	171	98	73	0	\	\	\	\	\	\	\	\	\	\	\	\
Jilesen	60	49	6	5	72	55	9	8	65	65	0	0	8	8	0	0	\	\	\	\
Han	\	\	\	\	94	52	42	0	\	\	\	\	\	\	\	\	\	\	\	\
Mihalache	10	7	3	0	8	8	0	0	\	\	\	\	\	\	\	\	\	\	\	\
Zhang	107	85	22	0	576	362	214	0	288	286	2	0	32	31	1	0	17	17	0	0
Sutton	13	3	10	0	184	75	98	11	77	54	14	9	\	\	\	\	8	7	1	1

**Table 4 jcm-13-03015-t004:** Pancreatic fistulae reported in all analyzed studies with the exclusion of Zerbi et al. [10,14,15,16,17,18,19,20,21,22,23,24].

	Pancreatic Fistulae											
Author Ref.	OS	CR	Lap	EN				DP				PD
				Overall	A	B	C	Overall	A	B	C	Overall
Fernandez-Cruz	\	\	\	8	4	3	1	2	2	\	\	\
Casadei	4	0	2	\	\	\	\	6	\	\	\	\
Zerbi	\	\	\	\	\	\	\	\	\	\	\	\
Zhao	112	9	11	\	\	\	\	\	\	\	\	\
Haugvik	\	\	\	7	1	6	0	7	0	7	0	\
Mehrabi	698	0	21	\	\	\	\	\	\	\	\	\
Xourafas	32	16	0	\	\	\	\	48	30	16	2	\
Jilesen	\	\	\	19	\	\	\	7	\	\	\	9
Han	26	0	29	\	\	\	\	2	43	11	1	\
Mihalache	\	\	\	2	\	\	\	2	\	\	\	\
Zhang	97	0	52	\	\	\	\	149	88	57	4	\
Sutton	20	0	\	\	\	\	\	\	\	\	\	\

Grades A, B, and C were reported only by 2 authors in the case of enucleation (EN) and by 5 authors in the case of distal pancreatectomy (DP). In the case of pancreaticoduodenectomies (PD), a fistula was reported overall by 1 author.

**Table 5 jcm-13-03015-t005:** The major post-surgical outcomes reported: length of hospital stay (LHS) and blood loss [10,14,15,16,17,18,19,20,21,22,23,24].

	LHS										BLOOD LOSS				
Author	Open			CR	Laparoscopy			EN	DP	PD	Open	Laparoscopy	EN	DP	PD
	Median	Max	Min		Median	Max	Min	Median	Median	Median	Median mL	Median mL	Median mL	Median mL	Overall
Fernandez-Cruz	\	\	\	\	\	\	\	5.5	6.7	\	\	<220	<220	\	\
Casadei	11	14	8	\	8	9.3	6.7	\	9.5	\	\	\	\	\	\
Zerbi	\	\	\	\	\	\	\	\	\	\	\	\	\	\	\
Zhao	21.2	38.5	3.9	27.8	15.1	22.9	7.9	\	\	\	163.6	124.8	\	\	\
Haugvik	\	\	\	\	7	27	2	8	6.5	\	\	300	500	300	\
Mehrabi	\	\	\	\	\	\	\	\	\	\	\	\	\	\	\
Xourafas	7	39	4	\	5	18	3	\	6	\	\	\	\	\	\
Jilesen	\	\	\	\	\	\	\	21	11	15	\	\	\	\	7
Han	9	66	7	\	7	18	4	\	8	\	\	\	\	\	\
Mihalache	\	\	\	\	\	\	\	\	\	\	\	\	\	\	\
Zhang	7	9	5	\	4	6	4	\	6	\	300	100	\	200	\
Sutton	\	\	\	\	\	\	\	\	\	\	\	\	\	\	\

CR: conversion rate; EN: enucleation; DP: distal pancreatectomy; PD: pancreaticoduodenectomy.

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
