# Peer review of "Pancreatic Neuroendocrine Tumors: What Is the Best Surgical Option?"

_jcm, 2024, doi:10.3390/jcm13103015_

Round 1
Reviewer 1 Report
Comments and Suggestions for Authors
Comments to Author
1. How did you ensure the consistency and reliability of data extraction across the twelve selected studies from different geographic regions and centers?
2. Can you elaborate on the criteria used to assess manuscript quality, and how did you address potential biases in the included studies?
3. Given the rarity of pNETs, did you encounter challenges in obtaining a sufficiently large sample size for your analysis?
4. Could you provide more details on the specific postoperative outcomes analyzed, such as the severity and management of pancreatic fistulae, postoperative bleeding, and re-operations?
5. Did your review uncover any significant differences in outcomes based on the type of minimally invasive approach used (e.g., laparoscopic vs. robotic)?
6. What factors were considered in determining the preference for enucleation over pancreaticoduodenectomy (PD) or distal pancreatectomy (DP) in certain cases, despite similar risks of postoperative pancreatic fistula (POPF)?
7. Were there any notable variations in surgical practices or outcomes among the countries or regions included in your study (Italy, the United States, China)?
8. How did you assess long-term follow-up and survival outcomes among patients undergoing different surgical interventions for pNETs?
9. What are your recommendations for improving multidisciplinary planning and patient selection for surgical management of pNETs based on your findings?
10. Considering the evolving landscape of minimally invasive techniques, do you foresee any changes or advancements in surgical approaches for pNETs in the near future?
Author Response
Dear Reviewer first of all, thank you for your time and for your appreciate work.
You can find in the attached file a point to point answer.
I'm sure, after your revision, that my paper earned quality and readability.
I hope the changes made may be sufficient.
- How did you ensure the consistency and reliability of data extraction across the twelve selected studies from different geographic regions and centers?
Thank you for your question. In our material and method we write: “The evaluation of manuscript quality was conducted using the Methodological Index for Non-Randomized Studies criteria [12] and the Newcastle-Ottawa Scale [13] to assess the quality of nonrandomized studies in meta-analyses because of the non-randomized nature of selected papers.”
We have carefully read all the selected papers. We used the NOS criteria e MINOR criteria to made a decision on quality of selected studies, but obviusly we can’t ensure the consistency and reliability of data presented in analyzed studis. But we add a specific sentence at the end of our discussion in order to better specify this limitations. You can find at 408-410 lines.
- Can you elaborate on the criteria used to assess manuscript quality, and how did you address potential biases in the included studies?
Thank you for your question. We try to clarify the difficulty to compare our selected studies results in discussion. In fact we underline, for every analyzed outcome, how many studies reported this results in order to clarify our difficulty. I hope I answer to your question.
- Given the rarity of pNETs, did you encounter challenges in obtaining a sufficiently large sample size for your analysis?
Thank you. Oh yes! This is one of the major problem. Only one study (Mehrabi et al.) reported a large sample. And this is one of most important reason of our paper.
- Could you provide more details on the specific postoperative outcomes analyzed, such as the severity and management of pancreatic fistulae, postoperative bleeding, and re-operations?
Yes we add here other table not used inside the main paper for a better readability. I hope is usefull for you.
- Did your review uncover any significant differences in outcomes based on the type of minimally invasive approach used (e.g., laparoscopic vs. robotic)?
Thank you. Unfortunately we have no data to drow conclusion. Sorry
- What factors were considered in determining the preference for enucleation over pancreaticoduodenectomy (PD) or distal pancreatectomy (DP) in certain cases, despite similar risks of postoperative pancreatic fistula (POPF)?
Thank you. Is not possible to summarize this data becouse of each Author, in his manuscript, consider different factor to prefer the surgical approach.
- Were there any notable variations in surgical practices or outcomes among the countries or regions included in your study (Italy, the United States, China)?
Thank you. The variability in the availability of the robotic platform must certainly be taken into consideration. Furthermore, the temporal differences in data collection certainly influenced the choice of surgical approach
- How did you assess long-term follow-up and survival outcomes among patients undergoing different surgical interventions for pNETs?
Thank you. We extraced data from selected paper and we underline in discussion all data found.
- What are your recommendations for improving multidisciplinary planning and patient selection for surgical management of pNETs based on your findings?
Thank you. We add a specific sentence at the end of conclusion. 424-428 line
- Considering the evolving landscape of minimally invasive techniques, do you foresee any changes or advancements in surgical approaches for pNETs in the near future?
Thank you. We add a specific sentence at the end of conclusion. 424-428 line
Kind regards
Reviewer 2 Report
Comments and Suggestions for Authors
Patreon et al in the review article titled ‘Pancreatic neuroendocrine tumors: what is the best surgical option?’ has discussed in detail about the characteristics, surgical practices, and postoperative outcomes describing the nature of pNET management. Enucleations and distal pancreatectomy were the most performed surgical techniques representing 43.4 and 38.6% of total interventions, respectively. Enucleation showed better outcome and postoperative outcomes compared to other techniques.
Lines 48-50: Please break the sentence and make two sentences. It’s too big to read.
Line 50: Please change ‘Its’ to ‘its’. There is capital letter in between the sentence.
Line 63: Please mention full form of ‘AIOM’
Lines 91-92: ‘The Authors considering the main Literature Collections of pNET patients evaluated current clinical’, please change the capital first letter to small letter for the bold words.
Lines 119-120: ‘We find a total of 3867 manuscripts for initial screening, using a systematic research. 1513 papers were excluded because of were duplicate’ – Please rewrite the sentence. Its not proper english.
Line 121: ‘Analyzing title, abstract and key-120 words, Authors selected the full-text version of 43 papers’. Please change the capital letter to small in bold word.
Line 144: Please mention the full form of ‘MINORS’
Line 184: Please keep one short form for pancreatic neuroendocrine tumor. There are two short forms in text pNET and PNET.
Line 205: Please change the capital first letter to small letter in ‘Literature’.
Line 257: Please check the spelling of parenchyma.
Line 268: Please mention full forms of ODP and LDP.
Lines 286 and 288: Please mention full forms of SDP.
Line 317: Please mention 14.6% and not 14,6%.
Comments on the Quality of English LanguageArticle needs extensive English editing. There are big sentences. There are capital letters used in between the sentences. Also there are many grammatical errors.
Author Response
Dear Reviewer. First of all, thank you for your time and for your appreciate work.
I’m Patrone and not Patreon but is not important i’m sure is an typing error.
I attach a new version of manuscript and a point by point answer.
I hope in this new form, our manuscript will be suitable of publication in your opinion.
----------------------------
Lines 48-50: Please break the sentence and make two sentences. It’s too big to read.
Thank you. I change the sentence. I hope it’s more clear.
Line 50: Please change ‘Its’ to ‘its’. There is capital letter in between the sentence.
Thank you. I change the sentence. I hope it’s more clear.
Line 63: Please mention full form of ‘AIOM’
Thank you. Changes made
Lines 91-92: ‘The Authors considering the main Literature Collections of pNET patients evaluated current clinical’, please change the capital first letter to small letter for the bold words.
Thank you. Changes made
Lines 119-120: ‘We find a total of 3867 manuscripts for initial screening, using a systematic research. 1513 papers were excluded because of were duplicate’ – Please rewrite the sentence. Its not proper english.
Thank you. We are so sorry. I change the sentence.
Line 121: ‘Analyzing title, abstract and key-120 words, Authors selected the full-text version of 43 papers’. Please change the capital letter to small in bold word.
Thank you. Changes made
Line 144: Please mention the full form of ‘MINORS’
Thank you. The full form was present in line 112
Line 184: Please keep one short form for pancreatic neuroendocrine tumor. There are two short forms in text pNET and PNET.
Thank you. Changes made
Line 205: Please change the capital first letter to small letter in ‘Literature’.
Thank you. Changes made. But, usually, when I cite the scientific literature I used the Capital letter. Maybe is a my mistake.
Line 257: Please check the spelling of parenchyma.
Thank you. I change the sentence
Line 268: Please mention full forms of ODP and LDP.
Thank you. Changes made
Lines 286 and 288: Please mention full forms of SDP.
Thank you. Changes made. A typing error
Line 317: Please mention 14.6% and not 14,6%.
Thank you. Changes made. A typing error